# Interhospital variation in the nonoperative management of acute cholecystitis

Konmal Ali[1], Nikhil L. Chervu[2], Sara Sakowitz[1], Syed Shahyan Bakhtiyar[3], Peyman Benharash[1,2]*, Shahin Mohseni[4], Jessica A. Keeley[5]

1 David Geffen School of Medicine, University of California, Los Angeles, Los Angeles, CA, United States of America, 2 Department of Surgery, David Geffen School of Medicine, University of California, Los Angeles, Los Angeles, CA, United States of America, 3 Department of Surgery, University of Colorado, Aurora, CO, United States of America, 4 Division of Trauma and Emergency Surgery, Department of Surgery, Örebro University Hospital, Örebro, Sweden, 5 Division of Trauma and Critical Care, Department of Surgery, Harbor-UCLA Medical Center, Los Angeles, CA, United States of America

* pbenharash@mednet.ucla.edu

**Data Availability Statement:** Data can be accessed at the Agency for Healthcare Research and Quality (https://www.hcup-us.ahrq.gov/) upon completion of Data Use Agreement for researchers who meet

## Abstract

### Background

Cholecystectomy remains the standard management for acute cholecystitis. Given that rates of nonoperative management have increased, we hypothesize the existence of significant hospital-level variability in operative rates. Thus, we characterized patients who were managed nonoperatively at normal and lower operative hospitals (>90th percentile).

### Methods

All adult admissions for acute cholecystitis were queried using the 2016–2019 Nationwide Readmissions Database. Centers were ranked by nonoperative rate using multi-level, mixed effects modeling. Hospitals in the top decile of nonoperative rate (>9.4%) were classified as Low Operative Hospitals (LOH; others:nLOH). Separate regression models were created to determine factors associated with nonoperative management at LOH and nLOH.

### Results

Of an estimated 418,545 patients, 9.9% were managed at 880 LOH. Multilevel modeling demonstrated that 20.6% of the variability was due to hospital factors alone. After adjustment, older age (Adjusted Odds Ratio [AOR] 1.02/year, 95% Confidence Interval [CI] 1.01–1.02) and public insurance (Medicare AOR 1.31, CI 1.21–1.43 and Medicaid AOR 1.43, CI 1.31–1.57; reference: Private Insurance) were associated with nonoperative management at LOH. These were similar at nLOH. At LOH, SNH status (AOR 1.17, CI 1.07–1.28) and small institution size (AOR 1.20, CI 1.09–1.34) were associated with increased odds of nonoperative management.

### Conclusion

We noted a significant variability in the interhospital variation of the nonoperative management of acute cholecystitis. Nevertheless, comparable clinical and socioeconomic factors

the criteria for access to confidential data. Data cannot be provided directly by the authors due to specific approval required by the Agency for Healthcare Research and Quality. The authors had no special access privileges to the data others would not have.

**Funding:** The author(s) received no specific funding for this work.

**Competing interests:** P. Benharash serves as a proctor for AtriCure. The content of this manuscript is unrelated to this work. The others authors have no disclosures. All authors declare no competing interests.

contribute to nonoperative management at both LOH and non-LOH. Directed strategies to address persistent non-clinical disparities are necessary to minimize deviation from standard protocol and ensure equitable care.

## Introduction

With a national annual incidence of 120,000, acute cholecystitis is responsible for an estimated $6.3 billion in healthcare expenditures [1, 2]. While cholecystectomy is the mainstay of treatment, rates of nonoperative management have increased over the last 40 years [3, 4]. Nonoperative therapies include antibiotics alone and image-guided procedures such as percutaneous gallbladder aspiration or drainage [5, 6]. Patient factors associated with non-operative management have been identified to include older age and public insurance [7, 8].

A growing body of literature has reported clinical success after nonoperative management of splenic injuries, appendicitis, and diverticulitis [9–11]. The decision to nonoperatively manage a patient is often due to comorbidity and age. However, resource availability and surgeon expertise may also influence the likelihood of nonoperative management [12]. Previous work in other surgical fields has suggested that tertiary hospital status is associated with higher rates of nonoperative treatment for low-risk thyroid cancers, small kidney masses, and splenic injuries [9, 10]. Although nonoperative management has proved efficacious in several disease categories, its utility in the case of acute cholecystitis has been debated with some reporting worse outcomes and increased complexity of eventual surgery [13]. Despite this, large-scale studies examining the center-level variation in the nonoperative management of acute cholecystitis are lacking.

In the present study, we used a nationally representative database to determine the presence of interhospital variation in the nonoperative management of acute cholecystitis. Furthermore, we compared the patient and hospital characteristics of nonoperatively managed individuals at lower operative hospitals to others. We hypothesized the presence of significant variability in operative rates attributable to hospital factors alone. Finally, while we predict advanced age and increased comorbid disease to reduce the likelihood of surgical management, socioeconomic and demographic factors may contribute to further disparities.

## Methods

### Data source and study population

We queried the 2016–2020 Nationwide Readmissions Database (NRD) for all adult ($\geq$ 18 years of age) admissions with a primary diagnosis of acute cholecystitis using *International Classification of Diseases codes, Tenth Revision* (ICD-10) codes. Maintained by the Healthcare Cost and Utilization Project, NRD is the largest all-payer readmissions database. Precise hospital discharge weights are used to account for 17 million annual discharges out of 36 million hospitalizations per year, accurately estimating 60% of all hospitalizations in the US from 28 states [14]. Patients with missing data for age, sex, or costs and those who underwent percutaneous cholecystectomy or were treated at hospitals with zero operative volume were excluded (3.6%; Fig 1).

### Variable definitions

Patient and hospital characteristics including age, sex, income quartile, primary payer, hospital setting, hospital teaching status, and bed size were defined according to the NRD data

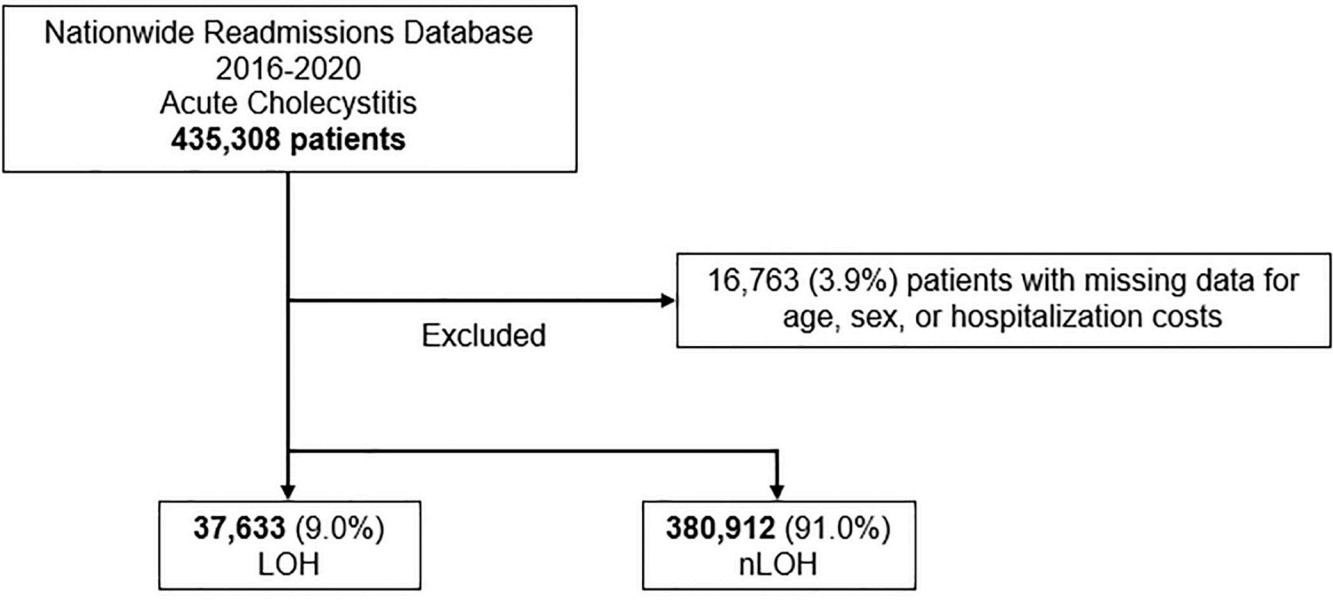

**Fig 1. Exclusion criteria; LOH, Low Operative Hospitals; nLOH, non-Low Operative Hospitals.**

dictionary. The van Walraven modification of the Elixhauser Comorbidity Index was used to incorporate the burden of chronic conditions into our analysis [15]. Additional patient comorbidities and complications were defined using ICD-10 codes. Patient comorbidities, complications, and additional procedures were determined using ICD 10 codes. The 2019 Personal Health Index was used to compute hospital prices, which were then adjusted for inflation using hospital-specific cost-to-charge ratios [16].

## Statistical analysis and study outcomes

We used multi-level, mixed-effects logistic regression modeling to rank hospitals by risk-adjusted nonoperative rates. Patient characteristics (age, sex, payer status, income level, and comorbid conditions) comprised the first level. Unique identifiers were used to treat hospitals as the second level. The "estat icc" command in Stata was then used to calculate the intraclass correlation coefficient (ICC). The ICC represents the proportion of total observed variation in the outcome due to interhospital differences. We subsequently generated baseline nonoperative rates at each center by estimating random intercepts via previously reported Bayesian methodology [17]. Hospitals with nonoperative rates $\geq 90^{th}$ percentile corresponding to a non-operative rate of 9.4% were classified as *Low Operative Hospitals* (LOH) with the rest classified as nLOH. Additionally, hospitals in the top quartile of Medicaid or self-pay (uninsured) admissions as defined by the Agency for Healthcare Research and Quality (AHRQ) were classified as safety net hospitals (SNHs).

Our primary outcome of interest was to determine if there was significant interhospital variation in the nonoperative management of acute cholecystitis independent of clinical and patient factors. We secondarily assessed patient, clinical, and hospital factors associated with nonoperative management at LOH and nLOH.

Continuous data are reported as means with standard deviation (SD) or medians with interquartile ranges (IQR) and categorical data as group proportions (%). The adjusted Wald and Pearson $\chi^2$ tests were used to determine the significance of intergroup differences amongst continuous and categorical variables, respectively. Relevant patient and hospital factors were selected for inclusion in regression models by Least Absolute Shrinkage Selection Operator

(LASSO) [18]. This regularization method improves out- of-sample model reliability and accuracy while reducing intervariable collinearity. Subsequently, two separate multivariable regression models were developed to determine factors associated with non-operative management at LOH and nLOH. Subgroup analysis was performed among the geriatric population (<65 years) to determine which factors were associated with nonoperative intervention after adjustment in this population. All models were adjusted for patient age, sex, and Elixhauser Comorbidity Index. Receiver operating characteristics (C-statistic) were used to optimize models in addition to Akaike and Bayesian information [19]. Regression outputs are presented as adjusted odds ratios (AOR) or beta-coefficients (β) with 95% confidence intervals (CI). An α less than 0.05 was set for significance.

Stata 16.1 was used for all analyses (StataCorp, College Station, TX). This study was deemed exempt from full review by the Institutional Review Board at the University of California, Los Angeles.

## Results

### Characteristics of operative and nonoperative patients

Of an estimated 418,545 patients who met study criteria, 83,272 (19.9%) were managed nonoperatively. Compared to operative patients, nonoperative patients were older (64.5 years ± 19.3 vs 54.6 ± 19.1), less commonly female (49.2 vs 58.3%), and had a higher burden of comorbid disease by Elixhauser Index (2 [3–5] vs 2 [1–3], all $p<0.001$). Nonoperative patients were more commonly in the lowest income quartile (29.3 vs 28.2%), and more likely to have Medicare (57.6 vs 37.0%), as opposed to private insurance (20.0 vs 34.5%, all $p<0.001$), compared to others. Patients managed nonoperatively were more frequently treated at SNH (26.2 vs 24.9, $p = 0.010$) and at metropolitan teaching institutions (70.9 vs 65.3%, $p<0.001$; Table 1).

### Interhospital variation in nonoperative management

A multivariable mixed-effects model was used to assess the impact of individual hospitals on variations in nonoperative management for acute cholecystitis. We found that 20.6% of nonoperative variation was attributable to interhospital differences by calculation of the ICC. Observed-to-expected ratios for each hospital were subsequently calculated with centers having a nonoperative rate > 9.2% classified as LOH (Fig 2).

### Characteristics of patients at Low Operative Hospitals

37,633 patients (9.9% of the entire cohort) were managed at 880 LOH. Compared to patients at nLOH, patients at LOH were similar in age (56.2 years ± 20.0 vs 56.6 ± 19.5, $p = 0.13$) and proportion female (56.0 vs 56.6%, $p = 0.24$). LOH patients were comparable in regards to income quartile distribution, but were less likely to have private insurance (30.1 vs 31.7%, $p<0.001$), compared to others. Regarding hospital characteristics, a higher proportion of LOH were designated as SNH (40.6 vs 27.4%) and they were more frequently metropolitan teaching institutions (56.4 vs 42.9%, both $p<0.001$), compared to nLOH. No difference in hospital size was noted between LOH and nLOH (Table 2).

After adjustment, increasing age (AOR 1.02/year, 95% CI 1.02–1.02), Medicare (AOR 1.31, 95% CI 1.21–1.43, ref: Private), Medicaid (AOR 1.43, 95% CI 1.31–1.57, ref: Private), and uninsured status (AOR 1.34, 95% CI 1.11–1.61, ref: Private) were among the factors associated with increased odds of nonoperative management at LOH (C-statistic: 0.68). Hospital factors such as SNH status (AOR 1.17, 95% CI 1.07–1.28) and small size (AOR 1.20, 95% CI 1.09–1.34, ref: Large) were also associated with higher odds of nonoperative management at LOH.

**Table 1. Baseline patient and hospital demographics of nonoperatively and operatively managed patients admitted for acute cholecystitis 2016–2019.**

| | Nonoperative (n = 83,272) | Operative (n = 335,273) | p-value |
|---|---|---|---|
| **Patient Characteristics** | | | |
| Age (years, mean ± SD) | 64.5 ± 19.3 | 54.6 ± 19.1 | <0.001 |
| Female (%) | 40,961 (49.2) | 195,553 (58.3) | <0.001 |
| *Income percentile (%)* | | | <0.001 |
| 0–25th | 24,112 (29.3) | 93,523 (28.2) | |
| 26th–50th | 22,605 (27.5) | 93,092 (28.1) | |
| 51st–75th | 19,456 (23.7) | 82,525 (24.9) | |
| 76th–100th | 16,024 (19.5) | 61,985 (18.7) | |
| *Payer (%)* | | | <0.001 |
| Private | 16,611 (20.0) | 115,256 (34.5) | |
| Medicare | 47,855 (57.6) | 123,794 (37.0) | |
| Medicaid | 12,737 (15.3) | 61,433 (18.4) | |
| Uninsured | 3,577 (4.3) | 21,044 (6.3) | |
| Other payer | 2,351 (2.8) | 13,074 (3.9) | |
| Elixhauser index (median, IQR) | 2 [3–5] | 2 [1–3] | <0.001 |
| *Comorbidities (%)* | | | |
| Cancer (metastatic) | 2,827 (3.4) | 2,048 (0.6) | <0.001 |
| Cancer (non-metastatic) | 5,254 (6.3) | 4,895 (1.5) | <0.001 |
| Cardiac arrhythmia | 21,050 (25.3) | 43,996 (13.1) | <0.001 |
| Congestive heart failure | 13,823 (16.6) | 18,249 (5.4) | <0.001 |
| Chronic lung disease | 16,028 (19.2) | 47,064 (14.0) | <0.001 |
| Chronic liver disease | 10,246 (12.3) | 33,669 (10.0) | <0.001 |
| Coagulopathy | 6,849 (8.2) | 11,979 (3.6) | <0.001 |
| Diabetes | 23,967 (28.8) | 66,949 (20.0) | <0.001 |
| End-stage renal disease | 15,209 (18.3) | 28,769 (8.2) | <0.001 |
| Hypertension | 52,966 (63.6) | 162,478 (48.5) | <0.001 |
| Neurological disorder | 5,957 (7.2) | 12,311 (3.7) | <0.001 |
| Obesity | 15,745 (18.9) | 86,558 (25.8) | <0.001 |
| Pulmonary circulatory disorder | 3,760 (4.5) | 5,304 (1.6) | <0.001 |
| Peripheral vascular disease | 7,380 (8.9) | 13,112 (3.9) | <0.001 |
| Rheumatologic disorder | 2,365 (2.8) | 7,291 (2.2) | <0.001 |
| **Hospital Characteristics** | | | |
| *Hospital status (%)* | | | <0.001 |
| Non-metropolitan | 6,836 (8.2) | 27,842 (8.2) | |
| Metropolitan non-teaching | 17,430 (20.9) | 88,691 (26.5) | |
| Metropolitan teaching | 59,006 (70.9) | 219,100 (65.3) | |
| *Bed Size (%)* | | | 0.73 |
| Large | 43,594 (52.4) | 176,828 (52.7) | |
| Medium | 24,176 (29.0) | 96,623 (28.8) | |
| Small | 15,503 (18.6) | 61,822 (18.4) | |
| Safety net hospital status (%) | 21,816 (26.2) | 83,520 (24.9) | 0.010 |

SD, standard deviation; IQR, interquartile range

Female sex (AOR 0.72, 95% CI 0.68–0.77, ref: male) and treatment at a metropolitan non-teaching (AOR 0.73, 95% CI 0.64–0.84, ref: non-metropolitan) or teaching institution (AOR 0.64, 95% CI 0.57–0.72, ref: non-metropolitan) were conversely associated with lower adjusted odds of nonoperative management (Fig 3).

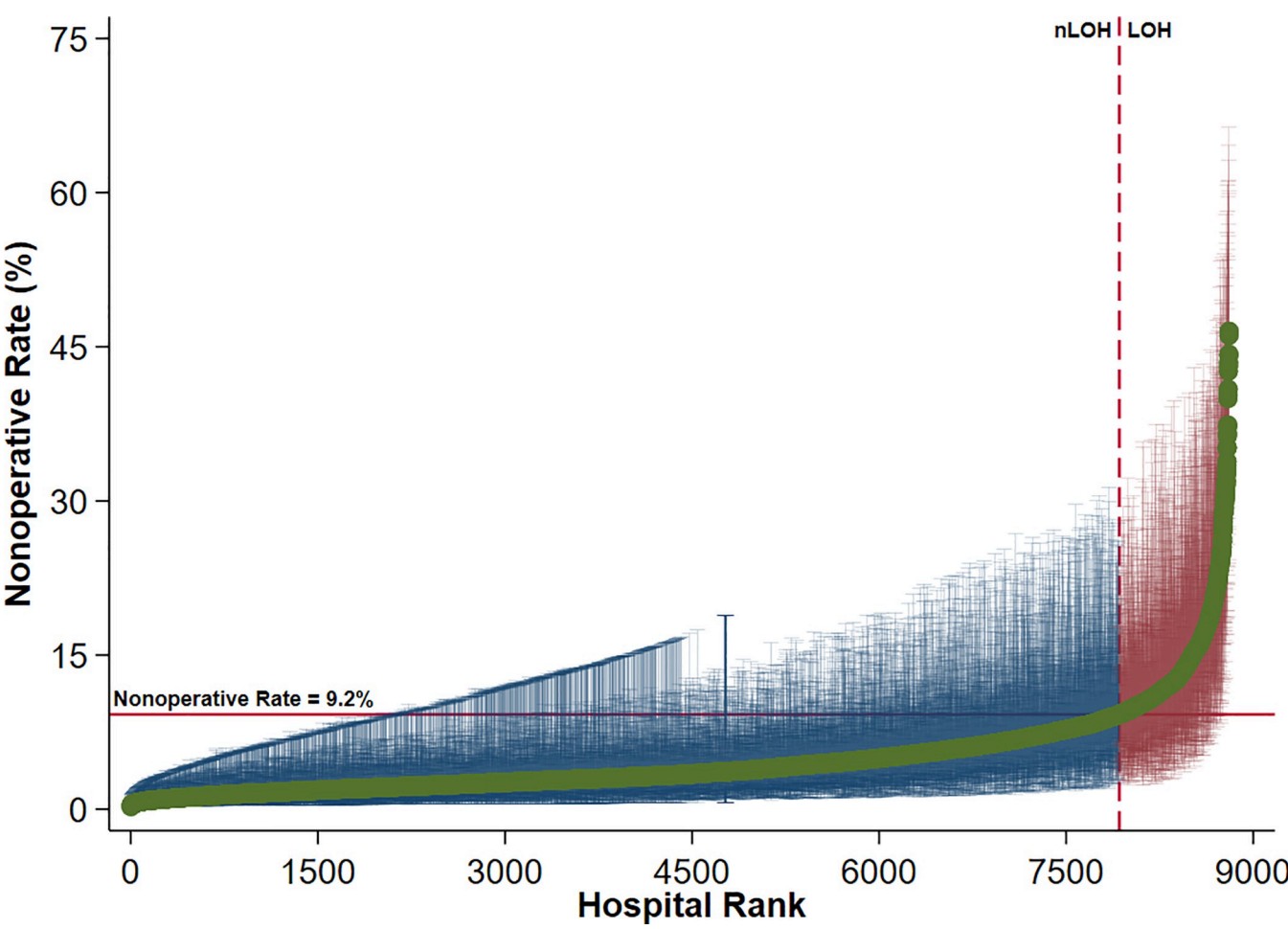

**Fig 2. Interhospital variation in nonoperative management of acute cholecystitis with 95% confidence intervals.** Centers in the top decile of nonoperative rate (>9.4%) were classified as Low Operative Hospitals; LOH, Low Operative Hospitals; nLOH, non-Low Operative Hospitals.

Multivariable analysis of nonoperative management at nLOH (C-Statistic: 0.72) demonstrated similar increased adjusted odds of nonoperative management associated with increasing age (AOR 1.02/year, 95% CI 1.02–1.03), Medicare (AOR 1.22, 95% CI 1.17–1.27, ref: Private), Medicaid (AOR 1.60, 95% CI 1.53–1.68, ref: Private), and uninsured status (AOR 1.53, 95% CI 1.43–1.64, ref: Private). Female sex (AOR 0.82, 95% CI 0.80–0.84, ref: male) remained associated with reduced adjusted odds of nonoperative management at nLOH. Importantly, nLOH patients treated at metropolitan teaching hospitals demonstrated increased adjusted odds of nonoperative management (AOR 1.25, 95% CI 1.18–1.34, ref: non-metropolitan). No association between hospital bed size or SNH status and adjusted odds of nonoperative management was observed at nLOH (Fig 3).

## Discussion

Although a growing number of general surgical diagnoses such as for appendicitis or diverticulitis are being managed without surgical intervention, objective criteria for nonoperative management is often sparse [9, 10]. In the present study, nearly 20% of patients underwent nonoperative management for acute cholecystitis. Likely due to the lack of standardized criteria, we noted the presence of significant interhospital variation. We found that LOH were

**Table 2. Baseline patient and hospital demographics of patients admitted for acute cholecystitis 2016–2019 receiving treatment at Low Operative Hospitals (LOH) versus non-Low Operative Hospitals (nLOH).**

| | LOH (n = 37,633) | nLOH (n = 380,912) | p-value |
|---|---|---|---|
| **Patient Characteristics** | | | |
| Age (years, mean ± SD) | 56.2 ± 20.0 | 56.6 ± 19.5 | 0.13 |
| Female (%) | 21,073 (56.0) | 215,441 (56.6) | 0.24 |
| *Income percentile (%)* | | | 0.061 |
| 0-25th percentile | 11,487 (30.8) | 106,148 (28.2) | |
| 26th-50th percentile | 10,139 (27.2) | 105,557 (28.1) | |
| 51st-75th percentile | 8,443 (22.6) | 93,567 (24.9) | |
| 76th-100th percentile | 7,228 (19.4) | 70,781 (18.8) | |
| *Payer (%)* | | | <0.001 |
| Private | 11,306 (30.1) | 120,731 (31.7) | |
| Medicare | 14,789 (39.4) | 156,861 (41.2) | |
| Medicaid | 8,584 (22.8) | 65,586 (17.2) | |
| Other payer | 1,178 (3.1) | 14,247 (3.7) | |
| Uninsured | 1,720 (4.6) | 22,901 (6.0) | |
| Elixhauser index (median, IQR) | 2 [1–4] | 2 [1–4] | <0.001 |
| *Comorbidities (%)* | | | |
| Neurological disorder | 1,523 (4.0) | 16,745 (4.4) | 0.029 |
| Cancer (metastatic) | 528 (1.4) | 4,347 (1.1) | 0.027 |
| Cancer (non-metastatic) | 1,097 (2.9) | 9,052 (2.4) | 0.002 |
| Cardiac arrythmia | 5,358 (14.2) | 59,689 (15.7) | <0.001 |
| Congestive heart failure | 2,670 (7.1) | 29,401 (7.7) | 0.028 |
| Chronic lung disease | 5,568 (14.8) | 57,523 (15.1) | 0.37 |
| Chronic liver disease | 3,365 (8.9) | 40,549 (10.6) | <0.001 |
| Coagulopathy | 1,566 (4.2) | 17,262 (4.5) | 0.054 |
| Diabetes | 8,241 (21.9) | 82,675 (21.8) | 0.63 |
| End-stage renal disease | 3,503 (9.3) | 39,332 (10.3) | 0.01 |
| Hypertension | 18,712 (49.7) | 196,731 (51.6) | 0.005 |
| Obesity | 8,650 (23.0) | 93,653 (24.6) | 0.004 |
| Peripheral vascular disease | 1,774 (4.7) | 18,717 (4.9) | 0.40 |
| Pulmonary circulatory disorder | 728 (1.9) | 8,337 (2.2) | 0.032 |
| Rheumatologic disorder | 777 (2.1) | 8,879 (2.3) | 0.029 |
| **Hospital Characteristics** | **n = 880** | **n = 7,927** | |
| *Hospital status (%)* | | | <0.001 |
| Non-metropolitan | 186 (21.1) | 1,924 (24.3) | |
| Metropolitan non-teaching | 198 (22.5) | 2,601 (32.8) | |
| Metropolitan teaching | 496 (56.4) | 3,402 (42.9) | |
| *Bed Size (%)* | | | 0.001 |
| Large | 302 (34.3) | 2,779 (35.1) | |
| Medium | 281 (31.9) | 2,486 (31.5) | |
| Small | 297 (33.8) | 2,652 (33.5) | |
| Safety net hospital status (%) | 357 (40.6) | 2,169 (27.4) | <0.001 |

SD, standard deviation; IQR, interquartile range

more commonly metropolitan teaching institutions and had a higher proportion of patients with non-private insurance. Despite these differences, factors such as increasing age, non-private insurance, and other socioeconomic characteristics were equally associated with adjusted

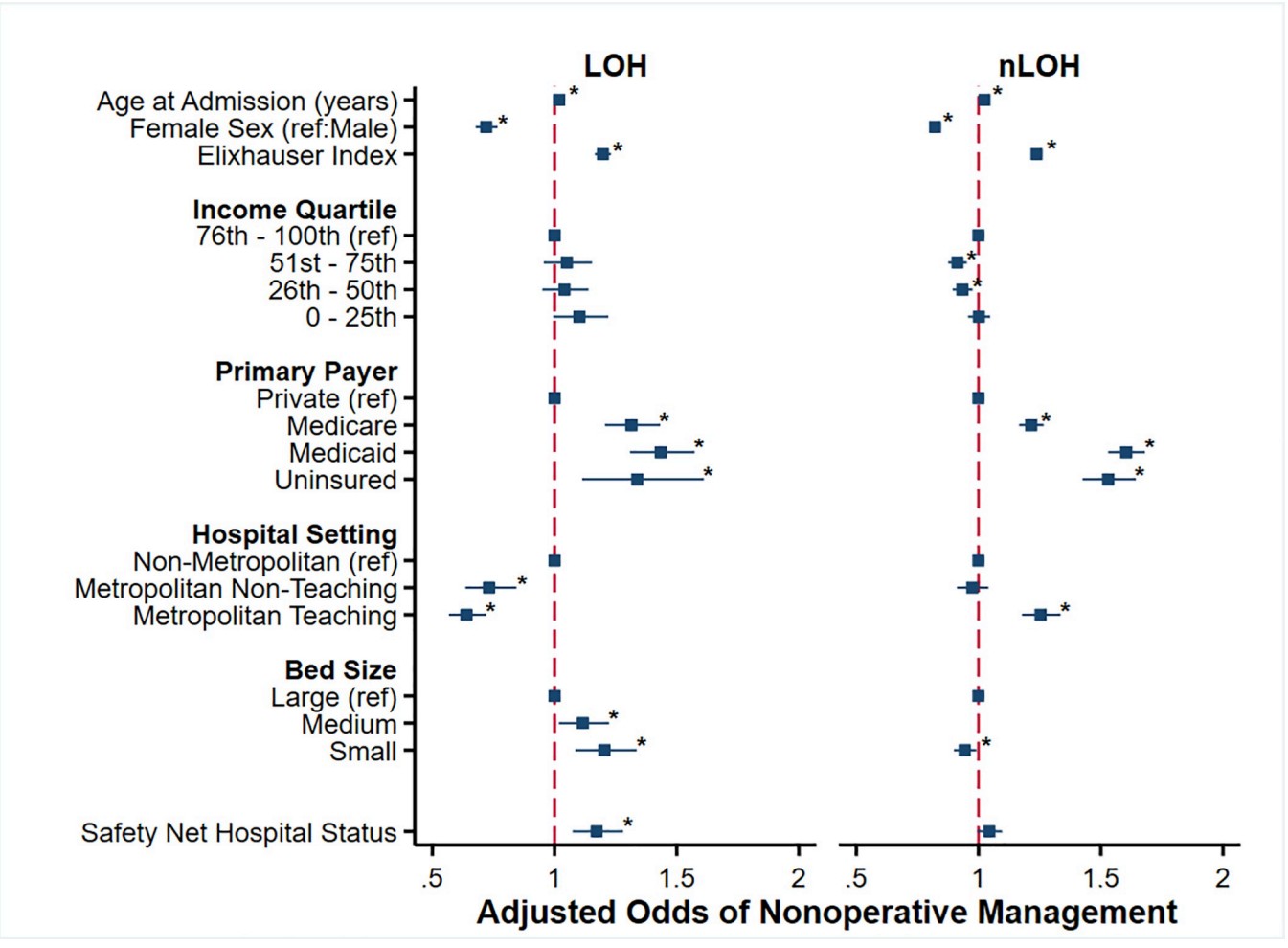

**Fig 3.** Factors associated with adjusted odds of nonoperative management for acute cholecystitis at Low Operative Hospitals (LOH; left) and non-Low Operative Hospitals (nLOH, right).

odds of nonoperative management at both LOH and nLOH. These disparities in operative management warrant further discussion.

Emphasis on value-based quality metrics has identified interhospital variation as a contemporary method to identify deviations from the standard of care [20]. Traditional measures, such as mortality and overall readmission rates, have been questioned due to inherent differences in patient populations, disease severity, comorbid conditions, and other socioeconomic factors [21, 22]. Accordingly, patient independent variation may indicate true quality differences between hospitals. After controlling for patient factors, we found that there was significant variation in the rate of nonoperative management for acute cholecystitis. Varying care practices can increase risk for inequitable application of nonoperative management and can result in increased overall complications. Particularly challenging cases notwithstanding, cholecystectomy is not usually resource-intensive or associated with significant morbidity [23, 24]. Prior studies have demonstrated that nonoperative management in poor surgical candidates is associated with a high rate of repeat admissions for recurrent cholecystitis [2]. Lack of symptom relief has been shown to result in > 30% readmission rates and the eventual need for cholecystectomy [2, 13, 25]. In addition, nonoperative management may worsen quality of life

as it often requires readmission or surgical intervention. A 2014 prospective randomized study found that patients undergoing delayed cholecystectomy had significantly lower patient satisfaction scores, compared to others [26]. Both persistent pain and need for readmission were primary factors in these lower scores. Given the variation in nonoperative rates attributable to hospital variation, communities may be subject to non-standard care based on nonoperative rates, suggesting that specific communities may be subject to non-standard care based on location, insurance status, and income.

LOH demonstrated significant differences in patient and hospital characteristics, relative to nLOH. Specifically, LOH cared for a greater proportion of patients with public insurance. Government-funded insurance and uninsured status have been associated with lower operative rates and overall inferior outcomes in the surgical oncology and orthopedic literature [27–29]. However, prior literature has reported conflicting findings regarding the association of insurance coverage with nonoperative management of emergency general surgical conditions [30–32]. In a study of ~140,000 patients presenting with acute cholecystitis, Loehrer et al found that expanded insurance coverage as part of healthcare reform was associated with increased rates of immediate cholecystectomy among self-pay or government-insured patients [32]. Yet, in another study, McCutcheon et al. reported no difference in rates of operative treatment among insured and uninsured patients [31]. Notably, LOH were also more often safety net hospitals. These hospitals play an essential role in caring for the underserved, but they may also face challenges in funding and staffing [33, 34]. While information assessing surgical team was unavailable for this analysis, future studies should investigate potential associations between surgeon specialty or expertise and non-operative management decisions.

Regardless of LOH status, increasing age and non-private insurance were associated with increased odds of nonoperative management in both cohorts. Prior studies have shown public insurance to be linked with nonoperative management. A 2021 study by Hashmi et al demonstrated that while privately insured patients made up 38.7% of operative EGS cases, they only made up 28.5% of overall admissions, signifying increased likelihood of surgical management compared to insured patients [35]. Similar decreases were found at the best-performing and worst-performing hospitals further indicating that patient insurance status is independently associated with nonoperative management. Although non-private insurance has been associated with increased disease complexity in EGS, this alone may not explain such differences [36]. Indeed, decreased access and fears regarding the ability to pay for surgical care have been cited as reasons for reduced operations in non-privately insured patients [37, 38]. Institutional and systems-based research examining patient and physician factors contributing to nonoperative cholecystitis management is necessary to address potential bias.

## Limitations

Our study has several limitations due to the use of an administrative database. The NRD does not possess granular data such as lab values or notation regarding specific physician or patient decision-making that may influence operative or nonoperative management. In addition, our study is reliant on accurate ICD coding, which is primarily used for reimbursements as opposed to future quality research. As a retrospective study, causal relationships cannot be made.

## Conclusion

In this nationwide retrospective study, we found significant interhospital variation in the operative rates for acute cholecystitis. We found that LOH were more likely to be metropolitan teaching hospitals with a higher proportion of non-privately insured patients. Regardless, we

found that older patients and non-privately insured patients were associated with higher adjusted odds of nonoperative management at both LOH and nLOH. Although improved antibiotics and new therapeutic modalities may augment nonoperative management, our observed variation likely points to insurance-based disparities. Future work should examine the factors affecting the choice to pursue nonoperative management at a more granular level to expose systemic biases.

## Author Contributions

**Conceptualization:** Konmal Ali, Nikhil L. Chervu, Syed Shahyan Bakhtiyar, Peyman Benharash, Shahin Mohseni, Jessica A. Keeley.

**Data curation:** Konmal Ali, Nikhil L. Chervu, Syed Shahyan Bakhtiyar, Jessica A. Keeley.

**Formal analysis:** Konmal Ali, Nikhil L. Chervu, Sara Sakowitz, Jessica A. Keeley.

**Investigation:** Konmal Ali, Nikhil L. Chervu, Sara Sakowitz, Jessica A. Keeley.

**Methodology:** Konmal Ali, Nikhil L. Chervu, Sara Sakowitz, Jessica A. Keeley.

**Project administration:** Konmal Ali, Nikhil L. Chervu, Jessica A. Keeley.

**Resources:** Konmal Ali.

**Supervision:** Nikhil L. Chervu, Peyman Benharash, Shahin Mohseni, Jessica A. Keeley.

**Validation:** Nikhil L. Chervu, Sara Sakowitz, Syed Shahyan Bakhtiyar, Peyman Benharash, Shahin Mohseni, Jessica A. Keeley.

**Visualization:** Konmal Ali, Nikhil L. Chervu, Peyman Benharash, Shahin Mohseni, Jessica A. Keeley.

**Writing – original draft:** Konmal Ali, Nikhil L. Chervu, Peyman Benharash, Shahin Mohseni, Jessica A. Keeley.

**Writing – review & editing:** Konmal Ali, Nikhil L. Chervu, Sara Sakowitz, Syed Shahyan Bakhtiyar, Peyman Benharash, Shahin Mohseni, Jessica A. Keeley.

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
