## [Decision Letter · Decision Letter 0]

31 Jan 2024

PONE-D-23-42781Interhospital Variation in the Nonoperative Management of Acute CholecystitisPLOS ONE

Dear Dr. Benharash,

Thank you for submitting your manuscript to PLOS ONE. After careful consideration, we feel that it has merit but does not fully meet PLOS ONE’s publication criteria as it currently stands. Therefore, we invite you to submit a revised version of the manuscript that addresses the points raised during the review process.

We look forward to receiving your revised manuscript.

Kind regards,

Barry Kweh

Academic Editor

PLOS ONE

Journal Requirements:

"Dr. Peyman Benharash received proctor fees from AtriCure as a surgical proctor. This manuscript does not discuss any products or services. Other authors report no conflicts."

We note that you received funding from a commercial source: AtriCure

Within this Competing Interests Statement, please confirm that this does not alter your adherence to all PLOS ONE policies on sharing data and materials by including the following statement: ""This does not alter our adherence to PLOS ONE policies on sharing data and materials.” (as detailed online in our guide for authors http://journals.plos.org/plosone/s/competing-interests).  If there are restrictions on sharing of data and/or materials, please state these. Please note that we cannot proceed with consideration of your article until this information has been declared. 

**Additional Editor Comments:**

A well conducted large study which requires a broader discussion to compare the authors' findings to the existing literature.

Reviewers' comments:

Reviewer's Responses to Questions

**Comments to the Author**

1. Is the manuscript technically sound, and do the data support the conclusions?

Reviewer #1: Yes

Reviewer #2: Yes

2. Has the statistical analysis been performed appropriately and rigorously? 

Reviewer #1: Yes

Reviewer #2: Yes

3. Have the authors made all data underlying the findings in their manuscript fully available?

Reviewer #1: Yes

Reviewer #2: No

4. Is the manuscript presented in an intelligible fashion and written in standard English?

Reviewer #1: Yes

Reviewer #2: Yes

5. Review Comments to the Author

Reviewer #1: 1. the topic is not unique but worthy of researching

2. there are many papers in google scholar and Refseek about this topic since 2020

3. Ethical approval is mentioned

4. the title is attractive

5. the abstract contains plagiarism. Otherwise it is good

6. the aim is clear

7. the KEYWORDS are good

8. lack of the abbreviations section

9. the introduction provide sufficient background information for readers in the immediate field to understand the problem/hypotheses

10. the text arrangement is good

11. the method section is good

12. the depth of the academic material is good

13. the study design is good

14. The suitability and accuracy of questions is good

15. The research methodology is clear

16. The materials are good

17. the logic is clear

18. the paper is novel

19. there are few grammatical errors in this article

20. the related concepts are introduced

21. the readability is sufficient

22. the results are good

23. all figures/tables are clear enough to summarize the results for presentation to the readers

24. all figures/tables are well referred to in the text

25. the theoretical analysis in this article is sufficient

26. the discussion of results from multiple angles is sufficient

27. the conclusion is tenable

28. the reference section contains too many old ref

29. please use (google scholar and Refseek) search engines then set it since 2019

30. the references are in order within the text

31. Bias is present

32. There is no conflict of interest with the author about this topic

33. Fund is mentioned

34. Conflict of interest is mentioned

35. Acknowledgement is mentioned

36. You can use my suggestions

My final decision is acceptable as it is

Reviewer #2: 1. Competing interests needs to be described according to the specific requirements of the journal.

2. PLOS Data Policy requires authors to explain the restrictions in detail (e.g., data contain potentially identifying or sensitive patient information) and provide contact information for a data access committee, ethics committee, or other institutional body to which data requests may be sent.

3. Reference: The referred literature is a bit outdated so cannot reflect the latest research trends. The authors need to describe more about the new research results rather than outdated research from the past.

4. Figures: Figures should be uploaded in TIFF or EPS format.

5. Tables: Place the title above the table and the legend or footnotes below the table.

6. Page 5 line 104: Delete the comma between “including” and “age”.

7. Methods: I would suggest adding subheadings and writing in sections to understand easier for readers.

8. Table 1,2：There’s no “clinical characteristics” in tables but at the title of the tables and the text.

9. Figure 2：Please explain how to definite the “hospital rank”.

10. Figure 3: The Pval for symbol “*” was not be identified.

6. PLOS authors have the option to publish the peer review history of their article (what does this mean?). If published, this will include your full peer review and any attached files.

Reviewer #1: **Yes: **hazim alhiti

Reviewer #2: No

---

## [Author Response · Author response to Decision Letter 0]

16 Feb 2024

Response to Reviewers 

Reviewer # 1:

1. the topic is not unique but worthy of researching

2. there are many papers in google scholar and Refseek about this topic since 2020

3. Ethical approval is mentioned

4. the title is attractive

5. the abstract contains plagiarism. Otherwise it is good

Thank you for your comment. We have made the following edits within the abstract (Page 2-3). 

 Cholecystectomy remains the standard management for acute cholecystitis. Given

that rates of nonoperative management have increased, we hypothesize the existence

of significant hospital-level variability in operative rates.

 All adult admissions for acute cholecystitis were queried using the 2016-2019

Nationwide Readmissions Database. Centers were ranked by nonoperative rate using

multi-level, mixed effects modeling.

We noted a significant variability in the interhospital variation of the nonoperative management of acute cholecystitis.

6. the aim is clear

7. the KEYWORDS are good

8. lack of the abbreviations section

9. the introduction provide sufficient background information for readers in the immediate field to understand the problem/hypotheses

10. the text arrangement is good

11. the method section is good

12. the depth of the academic material is good

13. the study design is good

14. The suitability and accuracy of questions is good

15. The research methodology is clear

16. The materials are good

17. the logic is clear

18. the paper is novel

19. there are few grammatical errors in this article

Thank you for your comment. We have fixed grammatical errors within the article. 

20. the related concepts are introduced

21. the readability is sufficient

22. the results are good

23. all figures/tables are clear enough to summarize the results for presentation to the readers

24. all figures/tables are well referred to in the text

25. the theoretical analysis in this article is sufficient

26. the discussion of results from multiple angles is sufficient

27. the conclusion is tenable

28. the reference section contains too many old ref

We appreciate your comment. We have made the following amendments to the references and have minimized the usage of any older references (Pages 16-21).

 [2] Escartín A, González M, Cuello E, et al. Acute cholecystitis in very elderly 

patients: disease management, outcomes, and risk factors for complications. Surg Res

Pract. 2019;2019:9709242. doi:10.1155/2019/9709242

[3] Parikh SS, Lindquester WS, Dhangana R. National cholecystostomy tube placement and cholecystectomy trends from 2010 to 2018. Journal of the American College of Radiology. 2023;20(6):537-539. doi:10.1016/j.jacr.2023.03.012

[5] Kurihara H, Binda C, Cimino MM, Manta R, Manfredi G, Anderloni A. Acute cholecystitis: Which flow-chart for the most appropriate management? Digestive and Liver Disease. 2023;55(9):1169-1177. doi:10.1016/j.dld.2023.02.005

[6] Kim SJ, Park HS, Lee DW. Outcome of nonoperative treatment for hip fractures in elderly patients: A systematic review of recent literature. J Orthop Surg (Hong Kong). 2020;28(2):230949902093684. doi:10.1177/2309499020936848

[11] Wells K, Fleshman J. Nonsurgical, minimally invasive, and surgical methods in management of acute diverticulitis. JAMA Surgery. 2019;154(2):172-173. doi:10.1001/jamasurg.2018.3112

[16] Using appropriate price indices for expenditure comparisons. Accessed February 15, 2024. https://meps.ahrq.gov/about_meps/Price_Index.shtml

[25] Mora-Guzmán I, Di Martino M, Bonito A, Jodra V, Hernández S, Martin-Perez E. Conservative management of gallstone disease in the elderly population: outcomes and recurrence. Scand J Surg. 2020;109(3):205-210. doi:10.1177/1457496919832147

[28] Haque LA. The effect of delays in acute medical treatment on total cost and potential ramifications due to the coronavirus pandemic. HPHR. 2021;26. Accessed February 15, 2024. https://www.jstor.org/stable/48617322

[33] Popescu I, Fingar KR, Cutler E, Guo J, Jiang HJ. Comparison of 3 safety-net hospital definitions and association with hospital characteristics. JAMA Network Open. 2019;2(8):e198577. doi:10.1001/jamanetworkopen.2019.8577

[34] Hefner JL, Hogan TH, Opoku-Agyeman W, Menachemi N. Defining safety net hospitals in the health services research literature: a systematic review and critical appraisal. BMC Health Services Research. 2021;21(1):278. doi:10.1186/s12913-021-06292-9

[38] Nath JB, Costigan S, Lin F, Vittinghoff E, Hsia RY. Access to federally qualified health centers and emergency department use among uninsured and medicaid‐insured adults: california, 2005 to 2013. Mycyk MB, ed. Academic Emergency Medicine. 2019;26(2):129-139. doi:10.1111/acem.13494

29. please use (google scholar and Refseek) search engines then set it since 2019

30. the references are in order within the text

31. Bias is present

32. There is no conflict of interest with the author about this topic

33. Fund is mentioned

34. Conflict of interest is mentioned

35. Acknowledgement is mentioned

36. You can use my suggestions

My final decision is acceptable as it is

Thank you for this comment. We have incorporated the above suggestions within our manuscript. 

Reviewer # 2:

1. Competing interests needs to be described according to the specific requirements of the journal.

Thank you for your comment. We have provided the following statement. 

P. Benharash serves as a proctor for AtriCure. The content of this manuscript is unrelated to this work. The others authors have no disclosures. All authors declare no competing interests. 

2. PLOS Data Policy requires authors to explain the restrictions in detail (e.g., data contain potentially identifying or sensitive patient information) and provide contact information for a data access committee, ethics committee, or other institutional body to which data requests may be sent.

We appreciate your comment. We have added the following statement with our submission. 

 The data utilized in the present study may be obtained directly from the Agency for

Healthcare Research and Quality (AHRQ), upon specific approval by the AHRQ for

data access. Users must also complete a mandatory Data Use Agreement in order to

access these data. For all researchers who meet these criteria, the NIS and all data of

the present study may be obtained directly from the AHRQ (website: www.hcup-

us.ahrq.gov; email: hcup@ahrq.gov). The authors of this work had no special

privileges to access the data that others would not have. Legally, we are not permitted to share the datasets directly, as part of the Data Use Agreement.

3. Reference: The referred literature is a bit outdated so cannot reflect the latest research trends. The authors need to describe more about the new research results rather than outdated research from the past.

Thank you for your comment. We have updated our reference list to ensure our work reflects the most current research trends. These changes can be found on (Pages 16-21).

 [2] Escartín A, González M, Cuello E, et al. Acute cholecystitis in very elderly 

patients: disease management, outcomes, and risk factors for complications. Surg Res

Pract. 2019;2019:9709242. doi:10.1155/2019/9709242

[3] Parikh SS, Lindquester WS, Dhangana R. National cholecystostomy tube placement and cholecystectomy trends from 2010 to 2018. Journal of the American College of Radiology. 2023;20(6):537-539. doi:10.1016/j.jacr.2023.03.012

[5] Kurihara H, Binda C, Cimino MM, Manta R, Manfredi G, Anderloni A. Acute cholecystitis: Which flow-chart for the most appropriate management? Digestive and Liver Disease. 2023;55(9):1169-1177. doi:10.1016/j.dld.2023.02.005

[6] Kim SJ, Park HS, Lee DW. Outcome of nonoperative treatment for hip fractures in elderly patients: A systematic review of recent literature. J Orthop Surg (Hong Kong). 2020;28(2):230949902093684. doi:10.1177/2309499020936848

[11] Wells K, Fleshman J. Nonsurgical, minimally invasive, and surgical methods in management of acute diverticulitis. JAMA Surgery. 2019;154(2):172-173. doi:10.1001/jamasurg.2018.3112

[16] Using appropriate price indices for expenditure comparisons. Accessed February 15, 2024. https://meps.ahrq.gov/about_meps/Price_Index.shtml

[25] Mora-Guzmán I, Di Martino M, Bonito A, Jodra V, Hernández S, Martin-Perez E. Conservative management of gallstone disease in the elderly population: outcomes and recurrence. Scand J Surg. 2020;109(3):205-210. doi:10.1177/1457496919832147

[28] Haque LA. The effect of delays in acute medical treatment on total cost and potential ramifications due to the coronavirus pandemic. HPHR. 2021;26. Accessed February 15, 2024. https://www.jstor.org/stable/48617322

[33] Popescu I, Fingar KR, Cutler E, Guo J, Jiang HJ. Comparison of 3 safety-net hospital definitions and association with hospital characteristics. JAMA Network Open. 2019;2(8):e198577. doi:10.1001/jamanetworkopen.2019.8577

[34] Hefner JL, Hogan TH, Opoku-Agyeman W, Menachemi N. Defining safety net hospitals in the health services research literature: a systematic review and critical appraisal. BMC Health Services Research. 2021;21(1):278. doi:10.1186/s12913-021-06292-9

[38] Nath JB, Costigan S, Lin F, Vittinghoff E, Hsia RY. Access to federally qualified health centers and emergency department use among uninsured and medicaid‐insured adults: california, 2005 to 2013. Mycyk MB, ed. Academic Emergency Medicine. 2019;26(2):129-139. doi:10.1111/acem.13494

4. Figures: Figures should be uploaded in TIFF or EPS format.

We appreciate your comment. All figures have been uploaded in TIF format. 

5. Tables: Place the title above the table and the legend or footnotes below the table.

Thank you for your comment. We have made the suggested changes in Table 1 and 2 (Pages 8-10). 

6. Page 5 line 104: Delete the comma between “including” and “age”.

We appreciate your comment. We have adjusted this error (Page 5). 

Patient and hospital characteristics including age, sex, income quartile, primary payer, hospital setting, hospital teaching status, and bed size were defined according to the NRD data dictionary.

7. Methods: I would suggest adding subheadings and writing in sections to understand easier for readers.

Thank you for your comment. We have added subheadings within the methods section to improve understanding for readers (Pages 5-7). 

8. Table 1,2：There’s no “clinical characteristics” in tables but at the title of the tables and the text.

We appreciate your comment. We have updated the titles of the tables to better reflect the data demonstrated (Page 8 and 10). 

Table 1: Baseline patient and hospital demographics of nonoperatively and operatively managed patients admitted for acute cholecystitis 2016-2019.

Table 2: Baseline patient and hospital demographics of patients admitted for acute cholecystitis 2016-2019 receiving treatment at Low Operative Hospitals (LOH) versus non-Low Operative Hospitals (nLOH).

9. Figure 2：Please explain how to definite the “hospital rank”.

Thank you for your comment. We have included information about the hospital rank within our figure caption (Page 9). 

 Figure 2: Interhospital variation in nonoperative management of acute cholecystitis

with 95% confidence intervals. Centers in the top decile of nonoperative rate (>9.4%) were classified as Low Operative Hospitals; LOH, Low Operative Hospitals; nLOH, non-Low Operative Hospitals.

10. Figure 3: The Pval for symbol “*” was not be identified.

We appreciate your comment. We have added the symbol to Figure 3 (Page 11).

---

## [Editor Report · Decision Letter 1]

6 Mar 2024

Interhospital Variation in the Nonoperative Management of Acute Cholecystitis

PONE-D-23-42781R1

Dear Dr. Benharash,

We’re pleased to inform you that your manuscript has been judged scientifically suitable for publication and will be formally accepted for publication once it meets all outstanding technical requirements.

Kind regards,

Barry Kweh

Academic Editor

PLOS ONE

Additional Editor Comments (optional):

The authors have strengthened by manuscript including grammatical errors and literature review following revisions suggested by the reviewers
---

## [Editor Report · Acceptance letter]

29 Apr 2024

PONE-D-23-42781R1 

PLOS ONE

Dear Dr. Benharash, 

I'm pleased to inform you that your manuscript has been deemed suitable for publication in PLOS ONE. Congratulations! Your manuscript is now being handed over to our production team.

Kind regards, 

on behalf of

Dr. Barry Kweh 

Academic Editor

PLOS ONE